Integumentary structure and composition in an exceptionally well-preserved hadrosaur (Dinosauria: Ornithischia)

Barbi Mauricio barbi@uregina.ca 1
Bell Phil R. 2
Fanti Federico 3 4
Dynes James J. 5
Kolaceke Anezka 1
Buttigieg Josef 6
Coulson Ian M. 7
Currie Philip J. 8
1 Department of Physics, University of Regina , Regina , Saskatchewan , Canada
2 School of Environmental and Rural Science, University of New England , Armidale , New South Wales , Australia
3 Dipartimento di Scienze Biologiche, Geologiche e Ambientali, Alma Mater Studiorum, Università di Bologna , Bologna , Italy
4 Museo Geologico Giovanni Capellini, Università di Bologna , Bologna , Italy
5 Canadian Light Source Inc., University of Saskatchewan , Saskatoon , Saskatchewan , Canada
6 Department of Biology, University of Regina , Regina , Saskatchewan , Canada
7 Department of Geology, University of Regina , Regina , Saskatchewan , Canada
8 Biological Sciences, University of Alberta , Edmonton , Alberta , Canada
Hutchinson John
Electronic publication date: 2019 Oct 16
Publication date: 2019
Volume: 7
Electronic Location ID: e7875
Received 2019 Apr 27; Accepted 2019 Sep 11
Copyright: ©2019 Barbi et al.
Copyright year: 2019
Copyright holder: Barbi et al.
License: This is an open access article distributed under the terms of the Creative Commons Attribution License, which permits unrestricted use, distribution, reproduction and adaptation in any medium and for any purpose provided that it is properly attributed. For attribution, the original author(s), title, publication source (PeerJ) and either DOI or URL of the article must be cited.
License URL: https://creativecommons.org/licenses/by/4.0/

Keywords: Integument, Preservation, Skin, Fossil, Hadrosaur, Cell layer, Synchrotron radiation, Scanning electron microscopy, X-ray, Spectromicroscopy

Funding: Australian Research Council DECRA Faculty of Sciences, University of Regina, Canada Canada Foundation for Innovation Natural Sciences and Engineering Research Council of Canada University of Saskatchewan Government of Saskatchewan Western Economic Diversification Canada National Research Council Canada Canadian Institutes of Health Research Phil R. Bell is funded by an Australian Research Council DECRA award and Mauricio Barbi by the Faculty of Sciences, University of Regina, Canada. Part of the research described in this paper was performed at the Canadian Light Source, which is supported by the Canada Foundation for Innovation, Natural Sciences and Engineering Research Council of Canada, the University of Saskatchewan, the Government of Saskatchewan, Western Economic Diversification Canada, the National Research Council Canada, and the Canadian Institutes of Health Research. There was no additional external funding received for this study. The funders had no role in study design, data collection and analysis, decision to publish, or preparation of the manuscript.

==============================
Preserved labile tissues (e.g., skin, muscle) in the fossil record of terrestrial vertebrates are increasingly becoming recognized as an important source of biological and taphonomic information. Here, we combine a variety of synchrotron radiation techniques with scanning electron and optical microscopy to elucidate the structure of 72 million-year-old squamous (scaly) skin from a hadrosaurid dinosaur from the Late Cretaceous of Alberta, Canada. Scanning electron and optical microscopy independently reveal that the three-dimensionally preserved scales are associated with a band of carbon-rich layers up to a total thickness of ∼75 microns, which is topographically and morphologically congruent with the stratum corneum in modern reptiles. Compositionally, this band deviates from that of the surrounding sedimentary matrix; Fourier-transform infrared spectroscopy and soft X-ray spectromicroscopy analyses indicate that carbon appears predominantly as carbonyl in the skin. The regions corresponding to the integumentary layers are distinctively enriched in iron compared to the sedimentary matrix and appear with kaolinite-rich laminae. These hosting carbonyl-rich layers are apparently composed of subcircular bodies resembling preserved cell structures. Each of these structures is encapsulated by calcite/vaterite, with iron predominantly concentrated at its center. The presence of iron, calcite/vaterite and kaolinite may, independently or collectively, have played important roles in the preservation of the layered structures.

Introduction

Fossilized dinosaur integument has been known for nearly 150 years, yet it is only recently that it has been considered more than a simple impression (i.e., trace fossil) of the original skin surface (Sternberg, 1953; Martill, 1991; Kellner, 1996). Although feathers and filamentary “protofeathers” of avian and non-avian theropods have received considerable attention, particularly in the past two decades, squamous (scaly) skin is more widespread and was probably plesiomorphic for Dinosauria (Barrett et al., 2015).

Significant advances in our understanding of the preservation and structure of squamous skin have been achieved with the use of synchrotron radiation techniques (Barbi, Tokaryk & Tolhurst, 2014 and references therein). Notably, it is now generally accepted that labile tissues, such as skin and muscle, can preserve and remain intact millions of years after the death of the organism (Schweitzer et al., 2005; Lingham-Soliar, 2008; Manning et al., 2009; Li et al., 2010; Zhang et al., 2010; Edwards et al., 2011; Li et al., 2012; Lindgren et al., 2014; Bertazzo et al., 2015; Lindgren et al., 2018).

Although hadrosaur skin is relatively common in the fossil record (Davis, 2014; Bell, 2014), few studies have investigated either its composition or the possible determining factors behind its preservation (e.g., Manning et al., 2009). We present a qualitative study of three-dimensionally preserved squamous skin (herein referred to simply as skin) from a hadrosaurid dinosaur from the Upper Cretaceous (Campanian) Wapiti Formation discovered near the city of Grande Prairie, Alberta, Canada. In this study, potential preserved structures that might be related to the original skin are investigated using a systematic approach employing scanning electron and optical microscopy, and an array of synchrotron radiation techniques. These techniques complement each other: scanning electron microscopy is used to map chemical elements and to identify different types of minerals in the sample; optical microscopy is used for morphological studies; synchrotron radiation techniques are employed to identify complex molecular structures and the chemical states of different elements along with their distribution in the sample, providing a suite of capabilities that goes beyond those attained with standard methods. It also requires less sample preparation than that required for optical and scanning electron microscopy. However, given the nature of the specimen under study, even synchrotron radiation techniques are not enough to fully determine, based on the observations, the original composition of the skin. It also lacks the capability to completely identify and understand the original sources of the chemical elements and, to some extent, their distributions in the specimen. This is mostly a consequence of the diagenetic modifications that the specimen has undergone. Nevertheless, the power of synchrotron radiation techniques used in this study resides in the fact that they can provide a series of results (for instance, the oxidation state of iron or the chemical state of carbon) that can be combined with other techniques to construct a likelihood (probabilistic) hypothesis to support a theory such as on the true nature or source of some of the structures observed in the hadrosaur skin. A more in-depth discussion on some of the limitations and advantages of synchrotron radiation techniques can be found in Hitchcock et al. (2008) and Yoon (2009).

Specimen and Geological Setting

UALVP 53290 represents an incomplete articulated-to-associated hadrosaurid skeleton comprising most of the thoracic region, forelimb and pelvic elements (Fig. 1). Parts of the tail likely continue into the cliff but could not be recovered owing to the precipitous nature of the outcrop. The only cranial element found, an incomplete jugal, indicates hadrosaurine affinities (triangular, asymmetrical anterior process; Horner, Weishampel & Forster, 2004); however, a second and more complete skeleton (UALVP 53722) found several hundred meters upstream of and approximately 10 m stratigraphically above UALVP 53290 is identifiable as Edmontosaurus regalis (Bell et al., 2014). As E. regalis is the sole hadrosaurine known from this stratigraphic interval (Bell & Campione, 2014), we tentatively refer UALVP 53290 to Edmontosaurus cf. regalis. Sheets of in situ and partially displaced fossilized integument were found close to the forelimbs of UALVP 53290 (Fig. 1A) and occur in two types: (1) as a 2 mm thick black rind preserving the three-dimensionality of the epidermal scales (Fig. 1B); and (2) as low-relief structures covered in a thin, oxide-rich patina (Fig. 1C). The skin samples examined here were slightly displaced relative to their true life position but are presumed to have come from the dorsal (anterior) surface of the forearm (area labelled “1” in Fig. 1A). The integument is composed of large (10 mm), hexagonal basement scales (sensu Bell (2012); type “1” according to the above descriptions) identical to scales on the upper surface of the forearm in other Edmontosaurus specimens (Osborn, 1912; Manning et al., 2009).

UALVP 53290 comes from the Upper Cretaceous deposits of the Wapiti Formation exposed near the city of Grande Prairie in west-central Alberta, western Canada. The specimen was collected at the Red Willow Falls locality, less than one kilometer to the east of the Alberta-British Columbia border.

Figure 1 Edmontosaurus cf. regalis (UALVP 53290).

(A) quarry map showing location of preserved integument (indicated by numerals) shown in (B) and (C). Dark grey regions are freshwater bivalves. (B) Higher magnification of 1 showing dark-coloured polygonal scales (type “1” preservation; see text). (C) Detail of 2 showing cluster areas associated with the forearm integument. (D) Detail of dark type “1” scales in oblique view showing sampling locations for spectromicroscopy: samples were collected with a microtome from (i) the outer surface of the epidermal scale to produce a light-coloured powder, and (ii) from a cross section of the scale that penetrated into the pale underlying sedimentary matrix to produce a dark-coloured powder. Scale bar in (A) is 10 cm. Scale bars in (B)–(D) are 1 cm. Abbreviations: Dv, dorsal vertebra; H, humerus; Mc, metacarpal; Os, ossified tendon; Pu, pubis; R, rib; Ra, radius; Sc, scapula; Th, theropod tibia; Ul, ulna. Line drawing by Phil R. Bell.

From a taphonomic perspective, the specimen comes from an undescribed monodominant hadrosaurine bonebed (sensu Eberth, Shannon & Noland (2007)), characterized by articulated-to-disarticulated elements of at least five individuals. Because exposures are on steep, sometimes overhanging cliffs, skeletal remains (including UALVP 53290 discussed here) are usually recovered in blocks derived from rock falls rendering precise mapping (including orientations, skeletal completeness and associations between individuals) virtually impossible. UALVP 53290 was partially disarticulated prior to burial, and curiously lacks vertebrae despite the fact that the ribs are still in approximate life position (Fig. 1A). Although the presence of skin would appear at odds with a period of subaerial exposure and disarticulation of the carcass, it is not without precedent, and instead provides circumstantial evidence of the structural integrity of the epidermis (see Herrero & Farke (2011) and Foster & Hunt-Foster (2011)). In this area, upper Campanian deposits of the Wapiti Formation have been dated at 72.58 ± 0.09 Ma (Bell, 2014) and consist of repeating fining-upward sequences of crevasse-splays, muddy and organic-rich overbank deposits, and minor sandy channel fills. Sandstones are primarily formed by poorly-sorted quartz, feldspar, and carbonate clasts, commonly presenting a carbonatic cement. Thin and discontinuous altered volcanic ash beds are found at the top of fining-upward successions, where they are locally interbedded with coal lenses (see also Fanti, 2009). Overall stacking pattern, facies distribution, and abundant megaplant material are indicative of a braided channel belt within a fluvial plain, characterized by high-water tables, poorly-drained soils, and abundant vegetation (Bell et al., 2013; Bell et al., 2014).

Paleogeographic reconstruction for the Campanian places the Red Willow Falls beds at approximately 400 Km north-west from the coeval paleoshoreline (Fanti & Catuneanu, 2009; Fanti & Catuneanu, 2010), and at a paleolatitude of approximately 65°N (Fanti & Miyashita, 2009 and references therein). Therefore, depositional conditions at the Red Willow Falls locality are unequivocally representative of terrestrial, freshwater conditions in a high-latitude setting.

In addition to the hadrosaur described here, this locality has yielded abundant cranial and postcranial hadrosaur elements referable to Edmontosaurus regalis, as well as isolated theropod (tyrannosaurid) teeth, diverse vertebrate tracks, and larval mayfly body fossils (Bell et al., 2013; Bell et al., 2014; Fanti, Bell & Sissons, 2013; Bell & Campione, 2014).

Materials and Methods

All fieldwork to collect UALVP 53290 was done with Palaeontological Collecting Permits issued by the Historical Resources Division (through the Royal Tyrrell Museum of Palaeontology) of the Government of Alberta, Canada. The permit to excavate Palaeontological Resources for the Grande Prairie area, file number 3950-605, was issued to Philip J. Currie, University of Alberta.

General strategy

As an overall strategy, and following an initial petrographic study of the sample, Scanning Electron Microscopy-Energy Dispersive X-ray spectroscopy (SEM-EDS) elemental maps of the skin were first derived to investigate possible markers that could (a) discriminate the skin from the sedimentary matrix, and (b) identify potential regions for the presence of organic contents. Subsequently, the chemical states of a series of elements were identified using X-ray methods, along with analyses using Fourier Transform Infrared Spectroscopy (FTIR).

SEM studies of a 20 µm thick cross section of the hadrosaur skin were conducted with the intentions listed. The thin section covers the first 2 cm of the sample starting from the outer surface of the scales (see section ‘Sample Preparation’). Characteristic X-ray lines, generated in processes involving the interactions between the electron beam and atoms in the sample, were used to produce elemental maps covering a ∼100 µm thick region spanning from the outer surface to the deepest regions of the sample, and single-point spectral measurements (see section ‘Results’ for details). To better understand the morphology of the hadrosaur skin, histological skin samples were prepared for a chicken (Gallus gallus domesticus), saltwater crocodile (Crocodylus porosus) and a rat (Rattus norvenicus) and compared with the hadrosaur skin.

Based on these observations, some samples of the skin (at least 100 µm thick and incorporating the outer surface of the scales) were tested at different beamline endstations at the Canadian Light Source (CLS), Saskatoon, Canada, using synchrotron radiation techniques. Only the data from two techniques are included in this paper, corresponding to Fourier Transform Infrared (FTIR) spectroscopy and X-ray Near Edge Spectroscopy (XANES). FTIR and XANES were used to probe for complex compounds and the chemical state of certain elements of interest, respectively. An elemental map using X-ray fluorescence techniques is included in the Supplemental Information (Section 1).

Sample preparation

Histological skin samples were prepared from skin that was dissected from the thigh (Gallus, Rattus) or foot (Crocodylus). Samples were preserved in 4% paraformaldehyde (PFA) for a three day storage period. Following storage, the samples were slowly introduced to 30% sucrose. Samples were then embedded in freezing media and sectioned in 20 µm thick slices along the transverse plane using a cryostat, allowing for identification of all skin layers. To better improve cell layer identification and contrast, skin samples were stained with eosin. This step was not required for that of the preserved hadrosaur skin, as sufficient contrast was already present in the sample.

Small (∼1 cm2) sections of pristine skin were sampled from the hadrosaur forearm during excavation to avoid contact with potential contaminants (anthropogenic or otherwise). Gloves were worn to collect the sample which were then wrapped in previously unused aluminum foils and placed in zipped plastic bags. The samples were kept in a storage room inside a clean drawer. Gloves were always used while manipulating the sample. All analytical samples were prepared from a single pristine skin sample.

For measurements using scanning electron microscopy and for the histological analysis, a 20 µm polished thin section was manufactured (Vancouver Petrographics, Langley, British Columbia, Canada) using standard petrographic thin section preparation techniques. Carbon was used to coat the sample (under high vacuum conditions) to improve conductivity before the SEM-EDS measurements. A small carbon signal is, then, expected in all spectra collected with this technique. Any excess carbon observed is assumed to be endogenous to the sample.

The CLS Mid-Infrared (MidIR) beamline endstation (May, Ellis & Reininger, 2007; Canadian Light Source, 2019a) was used for the FTIR measurements. Three different samples were prepared. Two of these were sampled from a pristine piece of the dinosaur skin still attached to the sedimentary matrix. A microtome (Microtome, Leica EM UC7) with a diamond blade was used to “scrape” one of the scales on two different surfaces (Fig. 1D): 1. on the outer (superficial) surface of the scale, producing a light-coloured powder (hereafter called “light-coloured powder”), and; 2. on the broken cross-section of the scale and incorporating some of the underlying sedimentary matrix, producing a dark-coloured powder (hereafter called “dark-coloured powder”). The third sample, also a small pristine piece of skin still embedded in sedimentary matrix, was immersed in HCl for two days to remove most of its mineral contents. The remaining sample was then washed with ethanol and air-dried at room temperature.

For XANES analysis, a piece of hadrosaur skin and the attached sedimentary matrix was sectioned at 100 nm into a water bath containing de-ionized water at room temperature (Microtome, Leica EM UC7). Due to the hardness of the sample, only flakes of the sample, containing skin and debris from the sediment, were obtained. The flakes and sediment were removed from the water bath using a wire loop and deposited onto a silicon nitride membrane, air-dried at room temperature and examined using X-ray spectromicroscopy with the Spectromicroscopy (SM) (Kaznacheyev et al., 2004; Kaznatcheev et al., 2007; Canadian Light Source, 2019b) beamline endstation.

Instrumentation

A Jeol JSM-6360 electron microscope (Department of Geology, University of Regina) and a Zeiss Axio Observer Z1 phase-contrast optical microscope (Department of Biology, University of Regina) were used to investigate the histological thin sections of the hadrosaur, chicken, rat, and crocodile skin. Linear measurements of various histological features were taken using the publicly available ImageJ 1.52k software.

A Nikon eclipse E600 Pol microscope (Department of Geology, University of Regina) was used for petrographic analysis of the hadrosaur skin and associated sedimentary matrix. Mineral identification were made utilising standard petrological microscope techniques; compositions were confirmed by reference to SEM-EDS analysis.

For the SEM measurements, the sample was subjected to high vacuum. To determine the best operational conditions, several areas were imaged at a range of beam currents (varied by increasing or decreasing the beam spot size), utilizing a smaller beam aperture and variable kV. Optimum conditions that provided the clearest and sharpest images at high magnification, but that did not lead to sample charging or sample degradation, were found to be 10–15 kV and a spot size of 65–70 µm. Examinations of samples were undertaken with both the secondary electron (SEI), and back-scattered electron (BSE) detectors fitted to the Jeol JSM-6360 instrument. A Noran 7 energy dispersive system attached to the Jeol SEM completed the X-ray spectroscopy analysis and mapping.

For FTIR measurements, the MidIR beamline provides a beam in the mid infrared region produced by a bending magnet, covering an energy range of 0.070–0.744 eV. A Hyperion 3000 IR microscope, cooled with liquid nitrogen, was used with the top objective set to 0.45 and the bottom objective set to 0.30, resulting in a beam spot size of 20 µm. 128 scans in transmission mode were performed per measured point. The data was processed using the Bruker commercial software package OPUS (Bruker, 2019) (baseline corrections were applied as provided with the software).

The scanning transmission X-ray microscope (STXM) endstation on the SM beamline (10ID-1) was used to collect all X-ray imagery and spectroscopy. Image sequences (i.e., stacks) (Dynes et al., 2006b) were collected at the C, Mg, Al and Si K-edges and at the K, Ca and Fe L-edges. The as-measured transmitted signatures (I) were converted to optical density (absorbance = OD = −In (I I0-1)) using the incident flux (I0) measured through areas where there was no sample present. The stacks (dimensions 8 × 8 µm) were collected at a spatial resolution of 60 nm and a dwell time of 1 ms/pixel. Principal component and cluster analysis (PCA-CA) was used to determine the number of chemical components present and/or derive spectra in selected image sequences (Lerotic et al., 2004). Quantitative maps from the image sequences were obtained using singular value decomposition (SVD) to fit the spectrum at each pixel to a linear combination of reference spectra of the components suspected to be present, or using spectra derived from the image sequence (Dynes et al., 2006a). Representative spectra were derived from the component maps using threshold masking (Dynes et al., 2006b; Dynes et al., 2006a). Axis2000 (Hitchock, 1997) was used for all STXM data processing.

The Very Sensitive Elemental and Structural Probe Employing Radiation from a Synchrotron (VESPERS) CLS beamline endstation (McIntyre et al., 2010; Canadian Light Source, 2019c) was also used to investigate possible chemical markers differentiating the skin from its associated sedimentary matrix. X-ray fluorescence spectroscopy using a polychromatic beam with energy ranging from 6 keV to 30 keV was employed, covering elements such as Ti, Ca, Mn, Fe, Zn, Sr, Y and Zr, among several others. The results of these measurements for Fe, Ca, Cu and Mn are presented with the Supplemental Information.

Results

Initial elemental mapping using the VESPERS beamline (Fig. S1) shows that the skin is enriched in Fe compared to the surrounding sedimentary matrix. In what follows, further results from measurements using SEM and SR techniques along with a histological analysis are presented.

Histology

Optical microscopy of histological samples of the skin from three extant representatives (Gallus, Crocodylus, Rattus) reveals a thin but characteristically multi-layered epidermis and a deeper, thicker dermis. The outermost epidermal layer corresponds to the stratum corneum, a keratinous layer that aids in protection of the internal organs (including the underlying epidermal and dermal layers) from desiccation. The stratum corneum, which is composed of stratified layers of β-keratin, is the thickest component of the epidermis in Crocodylus owing to the presence of keratinised scales (Fig. 2D). As a result of this cornified layer, the epidermis is also the thickest in Crocodylus in both absolute and relative terms. The stratum corneum is comparatively thin in both Rattus and Gallus, but where epidermal scales are present on the avian podotheca, such as Gallus, they are similarly covered by a thick stratum corneum (Pass, 1989), although these were not sampled in our specimen. In birds, the relative thickness of the epidermal layers differs between locations (Lucas & Stettenheim, 1972); however, the epidermis is consistently thinner in Gallus than it is in Rattus and Crocodylus, a feature that has been linked to the progressive lightening of the avian body and the evolution of flight (Stettenheim, 2000).

Figure 2 Comparative histology (transmitted light optical micrographs) of the skin of Edmontosaurus cf. regalis (UALVP 53290) (A, B), Crocodylus porosus (C, D), Rattus rattus (E, F), and Gallus gallus domesticus (G, H).

In UALVP 53290, the dark outer (superficial) layer corresponds to the position of the epidermis (e) in modern analogues (C–F). The thickness of the region identified as epidermis in UALVP 53290 varies (B); however, distinctive layering of this region (arrowheads in B) resembles the stratified appearance and general thickness of the stratum corneum in Crocodylus (D). Boxed area in A encompasses the enlarged area shown in B. (I) Phase-contrast and (J) transmitted light optical micrographs of Edmontosaurus cf. regalis (UALVP 53290) skin revealing fine laminae in the outer stratified region. The outermost epidermal layer in indicated by arrowheads. Dark laminae are, in places, composed of small, lenticular or subcircular bodies (arrows in I). Photos by Josef Buttigieg (A, B), Phil R. Bell (C–H) and Mauricio Barbi (I, J). Abbreviations: b, barite layer; e, epidermis; d, dermis; ds, dark stratified region; g, sedimentary grains; h, hinge area; hs, hair shaft; hy, hypodermis; m, sedimentary matrix; p, pigment cells; s, epidermal scale; sc, stratum corneum; sg, stratum germinativum. Silhouettes created by Pete Buchholz (Edmontosaurus, Buchholz (2019)), Rebecca Groom (Rattus, Groom (2019)), Steven Traver (Crocodilus, Gallus, Traver (2019a) and Traver (2019b)) courtesy of Phylopic and used under the Creative Commons Attribution-ShareAlike 3.0 Unported license (Commons, 2019).

In Crocodylus, the epidermis is invaginated to form the hinge area between scales (Fig. 2C). Much deeper invaginations, invading both the epidermis and dermis, are formed by hair and feather follicles in Rattus and Gallus, respectively (Fig. 2F).

Underlying the epidermis is the dermis, which contains openings for the blood vessels, fat deposits, and abundant pigment cells, the latter of which are more diverse in Crocodylus and other reptiles due to their naked skin (e.g., Bagnara & Hadley (1973); Fig. 2D). Glands are relatively scarce and/or small in avian and reptilian skin (e.g., Quay, 1972; Stettenheim, 2000; Jacobson, 2007), but are a salient feature of mammalian dermis (Quay, 1972; Fig. 2F). Thick subcutaneous hypodermis dominated by fat stores and blood vessels underlies the dermis in both Gallus and Rattus, whereas it is relatively thin in reptiles and not sampled in our specimen of Crocodylus.

Phase-contrast optical microscopy of the hadrosaur skin reveals an outermost (superficial) dark-coloured layer ∼35–75 µm in thickness, which overlies the sedimentary matrix (Figs. 2A and 2B). This outer layer is composed of clearly-defined, alternating dark and lighter-coloured layers, which typically range from ∼5 µm to ∼15 µm in thickness (Fig. 2B). Individual layers are typically laminar or undulatory giving the entire outer layer a stratified appearance. These finer layers may deviate around sedimentary particles that are occasionally found embedded within the outer dark layer (Fig. 2I). Aside from these occasional particles, sedimentary grains are typically restricted to the sedimentary matrix underlying the dark outer layer. In places, the dark layers appear to be composed of oval substructures measuring a few micrometers in maximum dimension (Fig. 2I). The entire dark stratified layer is, in places, capped by a pale-coloured, faintly laminated region identified as barite (see SEM and petrographic analyses below). No other evidence of integumentary features that could be interpreted as hair follicles, feathers or glandular structures could be identified anywhere in the sample. Other epidermal/dermal features such as osteoderms and melanosomes are also absent.

Petrography

The sample associated with the Hadrosaur skin is chiefly a fine to very fine grained detrital sedimentary rock (or subarkosic arenite), with local clay partings. The bulk of the grains are angular to subangular equant quartz (55–60%; 50–250 µm), and angular to subrounded clasts of K-feldspar/alkali-feldspar (10%; 25–100 µm), albite and plagioclase feldspar (5–10%; 50–150 µm). Chert/chalcedony is a minor component, with grains ranging from angular to subrounded (2–3%; 50–100 µm). Zircon (2–3%, 20–40 µm) is an important accessory phase, with many scattered angular and, more rarely, rounded grains occurring throughout the rock. Apatite, rutile and pyrite are also rare accessory minerals. Locally, euhedral grains of dolomite (2–3%, ∼100–150 µm) and barite (1%, up to 100 µm) occur, and may represent diagenetic phases. Calcite forms rare discrete grains (up to 50 µm), and is a common cement to the detrital grains (∼5%). Additional matrix minerals (<5%) include kaolinite and intermixed clays (potentially chlorite, smectite and illite), that commonly form local thin clay rich laminae, chiefly adjacent to the “skin” layer. The albite feldspar is commonly altered to sericite/muscovite and clay minerals.

Scanning Electron Microscopy (SEM) analysis

Back-scattered electron (BSE) SEM energy-dispersive X-ray imagery of a thin section comprising both skin and associated sedimentary matrix was used to identify concentrations of lower atomic number (low Z) elements (dark regions in Figs. 3C and 3D) and higher-Z elements (brighter areas in Figs. 3C and 3D). Thus, carbon-based structures (low Z), if present, are expected in the darker areas, while sedimentary material (high Z) will be mostly localized in brighter areas. SEM images show that, in general, the entire sample is dominated by angular particles of a sedimentary nature (white-to-grey in Figs. 3C and 3D, mostly detrital grains of quartz and feldspar). In contrast, the outermost 50–75 µm differs in both texture and concentration of elements. BSE SEM shows an outermost layer of white (high-Z; ∼25 µm thick) material identified as barite (Supplemental Information, Section 2) that caps a region of numerous thin, dark bands (low Z) up to a total thickness of ∼35–75 µm. The dark bands are interrupted by discontinuous laminae, or lenses, of higher Z material (grey in Fig. 3C).

To partially assess the elemental compositions of the light and dark bands, a number of spectral distributions (using SEM-EDS) were collected at different points on the sample as depicted in Fig. 3C. Most of these points (points 1 and 3–6 in Fig. 3C) were from grey or white areas (all high Z) identified by BSE SEM. Unsurprisingly, they showed no particular characteristic indicating contribution of organic components to each respective spectrum (Supplemental Information, Section 2). The most interesting spectrum comes from a region corresponding to lower-Z elements (point 2 in Fig. 3C and darker region indicated by an arrow in Fig. 3D). Figure 4 depicts this spectrum and is typical of all similar low Z areas distributed in the sample. These dark, carbon-rich areas are restricted to the outer parts of the sample and, in most cases, are covered with layers of kaolinite (among other minerals including quartz; point 5 in Fig. 3C; see also Supplemental Information). Interestingly, kaolinite appears to be mostly concentrated around these darker areas, which might indicate a correlation between carbon and kaolinite during diagenesis.

An elemental map of an area containing both high and low-Z regions (sedimentary and carbon-rich regions, respectively) was achieved using SEM X-ray in emission mode at a 5 µm scale. BSE imaging (Fig. 5A) of the carbon-rich (low Z) region confirms the identification of discrete layers or bands (dark structures in Fig. 5A). However, the SEM images alone are not sufficient to resolve those structures due to the two-dimensional nature of the measurement technique. Carbon distribution is directly correlated to the darker area in the sample (Fig. 5B), whereas oxygen, aluminum and silica are mostly concentrated outside the carbon-rich region (Figs. 5C, 5D and 5E, respectively).

Figure 3 SEM images from a sample of the skin from UALVP 53290 using (A, B) secondary electron (SEI), and (C, D) back-scattered electron detectors.

(A) Cross-section representing the epidermal scale and underlying sedimentary matrix (scale = 500 µm). Sedimentary deposits (e.g., individual detrital grains) are dominant in this image. (B) A magnified image of the boxed area shown in (A) (scale = 50 µm). The white top layer is of sedimentary nature (more details further below) and partially covers the top of several epidermal scales. A thin darker area under this white surface and above another sedimentary region can also be observed. (C) Higher magnification of boxed region in (B) (scale = 25 µm) imaged using BSE. Brighter areas represent higher atomic number (Z) elements. Areas richer in carbon-based (lower Z) structures are expected to show as darker regions, such as the one just under the top white layer in this image. Numbers and cross-hairs indicate specific points where spectral distributions were obtained using SEM-EDS. (D) BSE image showing a discrete darker zone (low Z; indicated with an arrow) close to the top of the sample (scale = 50 µm).

Figure 4 SEM chemical analysis for point 2 in Fig. 3C showing high carbon content typical of the dark (low Z) regions identified by BSE SEM.

Figure 5 SEM elemental maps for Carbon (B), Oxygen (C), Aluminum (D) and Silica (E) of a region containing a carbon-rich area and sediments.

The maps show a clear correlation between the darker area in the map (A) (BSE image) and the carbon distribution in the sample. Boxed area in (A) shows an apparent chain of sub-structures. The scale bar is 5 mµfor all figures.

Measurements with the MidIR beamline endstation

The following references were used for the identification of the different compounds in this section: Silverstein et al. (2014) for organic material; Derrick, Stulik & Landry (1999) for inorganic material. Extra references are provided in the text.

The first measurements using the MidIR beamline were carried out using the sample cleaned with HCl. This technique served to remove the hardest mineral parts from the sample leaving only a small amount of material associated with the epidermal scale. Most of this remaining material is identified as kaolinite (in agreement with the SEM results; Supplemental Information, Section 2), which is characterized by peaks at 3,690, 3,651, 3,619, 1,113, 1,026, 1,005, 936, and 910 cm−1 (Fig. 6) (Vahur, 2015; Madejová & Kamodel, 2001).

Figure 6 FTIR spectrum of the HCl-treated skin sample collected using Attenuated Total Reflectance (ATR) in absorption mode.

The peaks at 3,690, 3,651, 3,619, 1,113, 1,026, 1,005, 936, and 910 cm−1 are characteristics of kaolinite.

The peak at ∼1,635 cm−1 is likely related to the presence of montmorillonite (a type of smectite) and arises from the bending vibration of the OH group. The spectrum of montmorillonite has several peaks that superimpose with some others from kaolinite, although the peak at ∼1,635 cm−1 is strictly characteristic of the smectite group. On the other hand, the broad peak centered at ∼3,400 cm−1 is likely due to O-H stretch vibrations, which may be indicative of water impurities in the sample (Sigma-Aldrich, 2019). If that is true, O-H scissoring vibrational excitations could also explain the peak at ∼1,635 cm−1. Another possibility could be contamination by alcohol (used to clean the glass slides that hold the samples); however, the absence of other peaks characteristic of the alcohol group (Sigma-Aldrich, 2019), such as those at 1,420–1,330 cm−1, refutes this supposition.

Interestingly, the peaks located at ∼2,922 cm−1 and ∼2,849 cm−1 are likely the result of C-H vibrations and could provide evidence for the presence of organic material in the sample. The peak at ∼1,635 cm−1 could support this statement as it can be associated to the presence of amide-1 (indicative of carboxylic acid in the sample) (Sigma-Aldrich, 2019); however, as discussed above, this peak is likely of inorganic nature. Thus, the absence of other peaks that could support the presence of, for example, some carbonyl groups does not support the interpretation of an organic origin for those two peaks around 2,900 cm−1. Another plausible explanation for the appearance of those peaks is that they arise from C-O vibrations in carbonates. However, the complete absence of any significant peak near 1,400 cm−1 (characteristic of carbonates) makes this a weak proposition (Sigma-Aldrich, 2019).

The potential contamination by water absorption makes the above sample unsuitable for searching for organic material. The peaks arising from vibrational excitations of water molecules can superimpose those produced by vibrational modes in organic groups, making identification of the latter difficult.

Several maps and spectra were also collected from both the light and dark-coloured powder samples with the MidIR beamline using FTIR in transmission mode. The sample preparation in this case was less invasive (from a chemical perspective) and required less manipulation. While the light-coloured powder sample yielded spectra rich in clay mineral signatures (Supplemental Information, Section 4), such as kaolinite and montmorillonite, the dark-coloured powder provided a more suggestive selection of spectra.

Four spectra were selected from the map shown in Fig. 7 based on the features present in one or more of the areas of interest (Fig. 8) (the complete set of spectra can be found in the Fig. S7). While some minerals can be associated with some of the observed peaks, there are other peaks that likely indicate the presence of organic traces (Table 1).

Figure 7 Region of the darker powder sample mapped using FTIR in transmission mode.

Each cross on the map represents a position used to collect a spectrum.

Figure 8 Select set of spectra extracted from the map in Fig. 7 using FTIR in transmission mode.

A series of peaks identified using the OPUS software peak finder are listed on the top of the figure. A description of potential compounds associated to each of these peaks are provided in the text.

In particular, the appearance of the 1,789 cm−1 and 1869 cm−1 peaks in a FTIR spectrum is indicative of the presence of anhydrides acid (Coates, 2000). Interestingly, this acid is produced from the dehydration of carboxylic acids (Bruckner, 2010). The peak at 1,789 cm−1 corresponds to C-O symmetric stretch while that at 1,869 cm−1 corresponds to C-O asymmetric stretch.

Measurements using the SM beamline endstation

To further constrain the possible explanations for the observations described here, data were collected using the SM beamline from samples produced using a microtome. Optical images of the sample examined using the STXM are depicted in Fig. 9.

Carbon K-edge

The C K-edge image sequence was fit with protein, lipid and polysaccharide spectra (representing the major biomacromolecules commonly found in environmental samples), as well as carbonate and K (Dynes et al., 2006a). The spectra derived by threshold masking of these biomacromolecule component maps showed the same spectrum for each component map (Fig. 9A), which did not contain the characteristic peaks/spectral shape of protein (288.2 eV peak attributed to peptide band of proteins), polysaccharide (289.2 eV peak attributed to aliphatic C-OH) or lipids (Lawrence et al., 2003). The fact that only one spectrum could be derived indicates that the organic material was consistent throughout the skin. The main peaks in the spectrum are due to carbonyl (288.5 eV), ketone carbonyl (286.7 eV), carbonate (290.3 eV) and K (297.2 and 299.9 eV), C functional groups commonly found in soils (Gillespie et al., 2014).

Table 1 Associations between peaks in Fig. 8 and possible compounds (Silverstein et al., 2014; Derrick, Stulik & Landry, 1999).

Peak position	Possible association	
1,610 cm−1	Carboxylic acid, COO anti-symmetric stretch.	
1,680 cm−1	Carbonyl, C-O stretch α, β-unsaturated aldehydes and ketones; alkene, C-C stretch.	
1,789 cm−1 and 1,869 cm−1	Anhydrides acid, C-O symmetric stretch and C-O asymmetric stretch, respectively.	
2,514 cm−1	Calcite; Dolomite with calcite phase.	
2,849 cm−1 and 2,964 cm−1	Aliphatic compounds (alkenes), C-H vibrations.	

Figure 9 (A–C) Optical images of the hadrosaur skin and sediment debris from the microtome. The white rectangles show the region magnified in each successive image. (D) Transmission image at 280 eV of the boxed region in C, showing the inorganic material. The yellow rectangle shows the area studied in detail using STXM and depicted in Fig. 11. (E) Transmission image at 300 eV of the same area, showing the organic material.

In order to map the organic carbon, carbonate and K separately from each other, a modified carbon spectrum (referred to as “carbonyl”) was made by subtracting a carbonate spectrum (i.e., calcite) from it and removing the K peaks (Fig. 10B) (Dynes et al., 2006a). Two fittings were performed: one consisted of using calcium carbonate (i.e., calcite) and K (K2CO3 with carbonate signal subtracted) spectra, while the other used only a K2CO3 spectrum. Both fittings used the carbonyl spectrum and a slowly varying featureless signal (FS) (i.e., slightly decreasing straight line) (Dynes et al., 2006b), which represents the inorganic components present in the sample. The calcite and K spectra were published previously (Dynes et al., 2006a). This is the first time that the K2CO3 spectrum has been published to our knowledge. It exhibits the spectral shape of carbonates (Brandes, Wirick & Jacobsen, 2010). The component maps from the fittings are shown in Fig. 11. The carbonyl and FS maps are similar for both fittings, thus, only those from the first fitting are shown. It is apparent that the K2CO3 component map (Fig. 11D) is predominantly the sum of the carbonate and K component maps. Overlay of the carbonate and K component maps revealed that there are at least two different carbonate particles present: K2CO3 and another carbonate in the skin. In the sediment, only carbonates were identified (both calcite and dolomite were observed in the host rock, see section ‘Petrography’); neither organic carbon nor K was present Brandes, Wirick & Jacobsen (2010).

Figure 10 Carbon K-edge XANES reference spectra used in the initial linear regression fitting of the carbon image sequence and the organic carbon spectrum derived from the skin.

(A) Carbon spectrum derived from the skin using PCA-CA. Only one carbon spectrum is evident throughout the skin, consisting of peaks attributed to ketone carbonyl (286.7 eV), carbonyl (288.5 eV), carbonate (290.3 eV) and K (297.2, 299.9 eV). (B) To distinguish the organic carbon from the carbonate and K in the skin, the carbonate (calcite; red spectrum) and K peaks (green) were removed, resulting in a modified carbon in the skin spectrum (pink). To separate the carbonate from the K, the K2CO3 spectrum was modified by subtracting a calcium carbonate (CaCO3) spectrum from it. See text for details.

Figure 11 Carbon component maps of the area shown in Fig. 9E derived from the linear regression fitting of a C K-edge image sequence using reference spectra and the modified carbon (“carbonyl”) spectrum (see Fig. 10B).

(A) Carbonyl, (B) carbonate (CO3−2), (C) K, (D) K2CO3, and (E) featureless signal (FS). Two fittings were carried out; the first fitting used the modified carbon from skin spectrum, and the carbonate and K spectra. The second fitting used the K2CO3 instead of the K and carbonate spectra. Both fittings used the same slow varying featureless signal. The carbonyl and FS maps were similar for both fittings, thus, only those from the first fitting are shown. Color composites of selected component maps. (F) K, red; carbonyl, green; carbonate, blue; (G) K, red; FS, green; carbonate, blue; (H) K, red; K2CO3, green; CO3−2, blue.

Ca L-edge

Previous work (Benzerara et al., 2004; Fleet & Liu, 2009) has shown that the energy positions of the main Ca L-edge features are the same for the calcium carbonate polymorphs (e.g., calcite, aragonite, vaterite) and dolomite (CaMg(CO3)2 occurring at 249.3 eV (Ca LIII-edge) and 252.6 eV (Ca LII-edge). The number, energy positions, and relative intensities of the small peaks at lower energy relative to the main features are dependent on the coordination, reflecting electronic and crystal-chemical details within the first coordination sphere (Benzerara et al., 2004). Ca spectra derived from the image sequence from the skin and the sediment are shown in Fig. 12E. The ratio of the peak at 351.4 eV and adjacent to the main peak at 352.6 eV in the skin is 1.88. The ratio for the LII-edge peaks from the Ca in the sediment is 1.18. Hence, it appears that the Ca in the sedimentary areas occurs as calcite and in the skin as vaterite (Benzerara et al., 2004). However, potential spectral distortions associated with absorption saturation for optical density greater than two are possible, and may have decreased the ratio of these peaks, particularly for the Ca in the inorganic spectrum (Obst et al., 2009). Thus, it is possible that the CaCO3 in the sediment is vaterite and not calcite. Nevertheless, there is a small energy shift of the small peaks (Fig. 12F) from the skin and sediment indicating that their coordination of Ca is slightly different. Two types of Ca compounds are apparent when the Mg component map is overlaid with the Ca component maps (not shown); calcium carbonate and either dolomite or calcium carbonate containing significant amounts of Mg.

Figure 12 Calcium component maps derived from the linear regression fitting of a Ca L-edge image sequence using spectra derived from the Ca in the skin and in the inorganic particles (crystal) found associated with the integument.

(A) Ca in skin, (B) Ca in crystals, and (C) featureless signal (FS). (D) Color composites of the component maps (Ca in crystal = red, Ca in skin = green and slow varying featureless signal (FS) = blue). (E) Ca L-edge XANES spectra derived by threshold masking (Dynes et al., 2006b; Dynes et al., 2006a) of the pixels from the inorganic particles and from the skin component maps, (F) The spectra from the yellow box in (E) was enlarged and normalized on the 348 eV peak to better show the differences in peak position between the skin and inorganic spectra.

The Ca speciation in the sediment surrounding (and away from) the skin was also examined. Spectral fitting of the sediment Ca L-edge image sequence with the two spectra used in the skin/inorganic material fitting, and subsequent threshold masking of these component maps indicated that results similar to those for the skin were obtained, with regards to the ratio of the peaks. Thus it appears that absorption saturation is likely the reason for some of the peak ratio difference. In any event, only one species of Ca is apparent in the sediment, as the positions of the small peaks are the same between the two spectra. The Ca was observed to coincide with the carbonate map from the C K-edge fitting (Pearson’s coefficient 0.89), confirming that the Ca occurs as calcium carbonate.

Fe L-edge

The oxidation state of Fe can be determined from the relative intensities of the double peaked Fe L3 signal and from the position and number of peaks at the Fe L2–edge (Dynes et al., 2006b). The Fe L-edge image sequence for the skin was fit using two Fe reference spectra (Fig. 13A), representing Fe(III) (goethite) and Fe(II) (siderite) oxidation states and a featureless signal (FS) spectrum. The goethite and siderite spectra were previously published (Dynes et al., 2015). The Fe(III) and Fe(II) component maps are shown in Figs. 13B and 13C, respectively. The Fe(III) and Fe(II) spectra (3-point smoothed) derived from the respective component maps are characteristic of Fe(III) and Fe(II) spectra. The Fe(III) spectrum from the sample also contains considerable Fe(II). The Fe L-edge image sequence of the sediment was fit with the goethite, siderite and FS spectra. Threshold masking of these component maps indicated that there are Fe(II) and Fe(III) species in the sediments. The Fe(II) species in the sediment is similar to that found in the skin, whereas the Fe(III) species in the sediment contained more Fe(II) compared to that in the skin as evident from the equal heights of the two peaks at the Fe L3-edge. In agricultural soils, both Fe(III) and Fe(II) can coexist, even in well aerated soils, attributed to mixed valence Fe(II)-Fe(III) minerals and/or from periodic low-redox conditions where Fe(III) is reductively transformed to Fe(II)-bearing materials (Chen et al., 2014).

Figure 13 Iron component maps derived from the linear regression fitting of a Fe L-edge image sequence using reference spectra.

(A) Comparison of Fe(III) and Fe(II) spectra derived by threshold masking of the component maps to the siderite (Fe(II)) and goethite (Fe(III)) reference spectra. Component maps: (B) Fe(III), (C) Fe(II) and (D) slow varying featureless signal (FS). (E) Color composites of the component maps (Fe(III), red; featureless signal, green; Fe(II), blue).

Measurements of Al, Mg and Si K-edge were also performed and the data and discussions are provided with the Supplemental Information (Section 5).

Carbonyl, Ca and Fe maps of a carbon-rich layer and its substructures at the SM beamline

The skin flake was again measured at the SM beamline in order to identify the chemical state of the carbon associated with the optically-identified skin layers in Fig. 2. Measurements of the area containing the skin were performed with photon beam of 280 eV (corresponding to an energy lower than the carbon K-edge energy), and 288.5 eV (carbonyl K-edge excitation energy) using photon absorption mode. The image collected at 280 eV was used for background subtraction in the computation of the image at 288.5 eV. Red in Fig. 14A represents carbonyl after background subtraction, while cyan represents the measurements at 280 eV (non-carbonyl).

Figure 14 (A) Distribution of carbonyl (red) compared to the distribution of other compounds (cyan) in a 65 × 50 µm area of the sample. This area was measured in 0.1 µm steps. A layer of skin, identified by the yellow rectangle and predominantly composed of carbonyl, can be seen diagonally at the top-left corner of the image. The white arrow points towards the top of the skin. The red box represents the area detailed in (B). (B) Carbonyl (288.5 eV) map of the area within the red rectangle depicted in (A). The map covers an area of 20 × 20 µm of the sample in steps of 0.1 µm. The scale bar is 2 µm. Dark areas correspond to the presence of carbonyl (the data were collected with the detector in X-ray absorption mode). The yellow rectangle highlights a carbonyl layer which seems to be organized in smaller substructures, with three of them delineated by the white circles. (C) Elemental mapping of carbon (red), calcium (blue) and iron (green) of one of the substructures indicated by the white arrow in (B).

The yellow rectangle encapsulates a region of the sample with a layered structure similar to those identified in Figs. 2A, 2B, 2I and 2J. As Fig. 14 shows, this layer is rich in carbonyl, providing potential organic or organic-related material. The layers are clearly different both at chemical and morphological levels relative to other sedimentary regions of the sample.

A more detailed map of the area represented by the red rectangle in Fig. 14A was developed using a photon beam at 288.5 eV (carbonyl). This map is displayed in Fig. 14B. It shows potential carbonyl-rich substructures organized in a layer within the limits of the dinosaur skin. The existence of these substructures is reinforced by the evidence observed in both the optical and SEM images of the skin. The images captured using the three different techniques (X-ray SR, SEM and optical) all show similar geometrical features with apparent wave-like contours to the skin layers. These characteristics strongly suggest organized structures composed of subcircular substructures resembling cells. Artifacts introduced by the measurements can be discarded with high likelihood as it is unlikely that three completely different and independent techniques would randomly yield similar geometric features.

One of the substructures was mapped for carbon, Ca and Fe (Fig. 14C). As expected, the chemical states of these elements were found similar to those observed in section ‘Measurements using the SM Beamline Endstation’ for the carbon-rich area (predominantly carbonyl, vaterite/calcite and goethite). As depicted in Fig. 14C, carbonyl is relatively uniformly distributed, while vaterite/calcium appears to delineate the substructure. More remarkably, goethite is highly concentrated at the center of this form. We have not investigated the reasons behind these distributions, and can only speculate about them at this point.

Discussion

Histological sampling of the hadrosaur skin reveals microscopic details of the dark outer layer associated with a single epidermal scale. Specifically, this layer is distinctly stratified, composed of alternating dark and lighter-coloured layers with a total thickness of ∼75 µm. The topological position, overall thickness and stratified composition of the dark outer layer in UALVP 53290 is strongly reminiscent of the stratum corneum in Crocodylus (∼145 µm thick in Crocodylus), which forms the thickest component of the epidermis in the latter. In contrast, the entire epidermis is extremely thin in both Rattus and Gallus (< 25 µm) and the thickness of the stratum corneum is negligible compared to Crocodylus. Given the obviously scaly epidermal covering of hadrosaurs (Bell, 2014), including UALVP 53290, it seems reasonable to infer that the dark-coloured stratified layer represents the mineralized remains of the stratum corneum. The differing thickness in what we have identified as the stratum corneum of the hadrosaur and that of Crocodylus could be attributable to dehydration (Chang et al., 2006; Papageorgopoulou et al., 2015), diagenesis, taxonomic differences or any combination of these. Similar keratinous structures to those identified in UALVP 53290, together with intact remains of α- and β-keratins have been reported in non-avian dinosaurs and contemporaneous birds (Schweitzer et al., 1999a, Schweitzer et al., 1999b; Schweitzer, 2011; Pan et al., 2016); however, these results are largely restricted to feathers and the cornified sheaths covering the unguals rather than skin.

Measurements using the MidIR and SM beamlines have provided evidence of organic components in the mineralized skin. Both beamlines demonstrated that carbon in the carbonyl form is associated with the carbon-rich area observed with SEM measurements. The true nature and origin of such organic (or organic-related) matter could not be determined.

Measurements of an individual substructure in one of these layers (Fig. 14C) show conspicuous evidence apparently unrelated to random processes induced by diagenetic modifications, but rather, driven by the skin architecture. The associated organic material might, or might not, be original to the animal. In any case, the observed organic and inorganic materials are likely associated with the diagenetic processes that led to the preservation of the skin.

We also found that iron, in the form of FeCO3, is predominantly found at the center of one of these substructures forming the skin layers (Fig. 14C). These substructures closely resemble the geometry of skin cells that have been delineated and filled with vaterite/calcite.

It is also interesting to note that unlike previous suggestions where carbonate was implicated in the preservation of soft tissues (Manning et al., 2009), the skin layers reported here are predominantly associated with carbon in the form of carbonyl (Fig. 14B). Therefore, the conditions that led to the preservation of soft tissues in UALVP 53290 were likely very different from those reported by Manning et al. (2009). These conditions likely involved the early presence of kaolinite, which shrouded parts of the integument and provided an effective barrier to decay (Naimark et al., 2016b; Naimark et al., 2016a; McMahon et al., 2016), and the subsequent chemical reactions perhaps related to post-mortem bacterial activities (the organic material may be related to these activities), which is supported by the potential presence of anhydride acid (Table 1 and associated discussions). Hence, it is reasonable to infer that a combination of factors including the presence of kaolinite, organic-related material, FeCO3 and vaterite/calcite contributed to the preservation of epidermal layers in UALVP 53290.

In summary, following the identification of low Z regions in association with the epidermal scales (based on SEM analysis), several interesting features have been observed. Examinations based on correlation analysis between different peaks observed at the MidIR beamline provided further insight into the sample content. However, and based on the full dataset and results collected with the SM and MidIR beamlines, SEM and optical microscope, the message is that the sample contains a complex set of compounds, with non-negligible evidence for the existence of organic-related material of some nature still associated with the epidermal scales.

The association of iron and kaolinite with the skin might indicate part of the diagenetic process involved in the preservation of this specimen. For example, kaolin might have accumulated over the surroundings of the animal after its death as part of the fossilization process. Assuming the darker areas in the thin-section consist of the permineralized epidermis, we can speculate that the presence of kaolin might have produced an extrinsic barrier to decay and/or promoted the preservation of labile tissues; this is in part due to the small grain size and high charge of clay particles inactivating bacterial degradative enzymes (Naimark et al., 2016a). On the other hand, the abundance of iron in the oxidation state Fe(III) (goethite) was found to differ between the skin and associated sediments (see section ‘Fe L-edge’). The data collected with the SM beamline endstation supports a strong presence of goethite associated with the skin. As suggested by Schweitzer et al. (2013), iron in this state may play a role in the preservation of soft tissue by stabilizing and making it more likely to be preserved.

The depositional environment into which UALVP 53290 was buried unquestionably played a fundamental role in the early mineralization of integumentary tissues. Sedimentological observations at the Red Willow falls outcrop combined with petrographic analyses confirm: 1. a fresh-water setting; 2. the presence of abundant plant material, and 3. high hydraulic energy and sediment supply into the system (Fanti, Bell & Sissons, 2013). Most importantly, UALVP 53290 stands amongst the best examples of “mummified” dinosaurs not preserved in distal alluvial plain deposits, where brackish to peatland environments may have influenced the precipitation of minerals around and within tissues (Martill, 1991; Manning et al., 2009).

Conclusion

The main outcome of this work is the structural and compositional description of layers compatible with epidermal cell layers in the skin of a hadrosaur. These layers, which are topographically and morphologically congruent with the outer, cornified layer of the epidermis (stratum corneum) of modern reptiles, are, in places, composed of chains of carbon-rich subcircular bodies, predominantly in the form of carbonyl. Detailed analyses integrating different measurement techniques in synchrotron radiation, scanning electron microscopy and optical microscopy documented that calcite encapsulates these substructures, with iron (in the form of goethite) concentrated at its center. The source and distribution of iron in the substructures remains unclear, but which most likely played a role, along with vaterite/calcite, in the preservation of the micrometric structures. Kaolinite also appears to have contributed to the preservation of the skin structure by encapsulating it and retarding organic decay.

The complex methodology applied in this study resulted in the following procedure:

1. fossilized skin tissue preserving carbon-rich layers were observed using scanning electron microscopy (SEM);

2. Measurements with synchrotron radiation in the mid-infrared (mid-IR) region of the spectrum revealed evidence of carbon in different organic forms, with carbonyl associated to the carbon-rich layers;

3. Synchrotron X-ray spectromicroscopy (SM) corroborated the previous observations by producing a map of the distribution of carbonyl. The distribution showed a strong correlation between carbonyl and the carbon-rich layers, establishing that carbon in the skin is predominantly of organic nature.

4. Optical microscopy along with SEM and SR images indicate that carbonyl layers are topologically organized and composed of smaller, subcircular substructures (Figs. 2, 5 and 14). The likelihood that these substructures are the product of geometrical or optical effects introduced in the measurements, or just random distributions, is minimized by the fact that the three techniques used are completely independent of each other, yet they returned similar results.

This study, thus, represents an attempt to combine different cutting-edge optical and chemical analyses. Future quantitative analysis and histological studies, however, should be directed towards addressing how these structures were preserved and how they compare to related extant analogues (e.g., crocodylians, birds).

Supplemental Information

Supplemental Information 1 Supplemental material

Click here for additional data file.

UALVP 5320 was discovered and prepared by Phil R. Bell and Robin Sissons. We thank all the students and volunteers, especially Jeff Brewster for assistance collecting UALVP 53290. Grande Prairie Regional College, Eva Koppelhus (University of Alberta), Dick Barendregt and Certek Heat Machine are thanked for logistical assistance while in the field. We thank Renfei Feng and Ferenc Borondics for the technical support with the measurements at the VESPERS and MidIR beamlines at the Canadian Light Source, respectively.

Additional Information and Declarations

Competing Interests

Author Contributions

Field Study Permissions

Data Availability

James J. Dynes is employed by Canadian Light Source Inc.

Mauricio Barbi, Phil R. Bell, James J. Dynes and Anezka Kolaceke conceived and designed the experiments, performed the experiments, analyzed the data, contributed reagents/materials/analysis tools, prepared figures and/or tables, authored or reviewed drafts of the paper, approved the final draft.

Federico Fanti and Ian M. Coulson conceived and designed the experiments, performed the experiments, analyzed the data, contributed reagents/materials/analysis tools, authored or reviewed drafts of the paper, approved the final draft.

Josef Buttigieg conceived and designed the experiments, performed the experiments, contributed reagents/materials/analysis tools, prepared figures and/or tables, authored or reviewed drafts of the paper, approved the final draft.

Philip J. Currie authored or reviewed drafts of the paper, approved the final draft.

The following information was supplied relating to field study approvals (i.e., approving body and any reference numbers):

Field experiments were approved and Collecting Permits were issued by the Historical Resources Division (through the Royal Tyrrell Museum of Palaeontology) of the Government of Alberta, Canada (file # 3950-605).

The following information was supplied regarding data availability:

The data are available as histograms and tables in the article and Supplemental Information.

All data are presented as spectral histograms. Chemical maps in this work are qualitative rather than quantitative, with no importance given to absolute values. In fact, the nature of this work is fundamentally qualitative.

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
