# Peer review of "Integumentary structure and composition in an exceptionally well-preserved hadrosaur (Dinosauria: Ornithischia)"

_PeerJ, doi:10.7717/peerj.7875_

## Round 0.1 · original submission · Major Revisions

Thank you for submitting this intriguing and rigorous manuscript. As you will see, the two reviewers find it as valuable as I do and they give plenty of constructive critiques about the wording, detail and interpretations. Together, these constitute moderate revisions. Notably, both commend the authors for not over-interpreting the data, in an area of palaeobiology that some might consider to be too over-interpretive or at least controversial. I applaud that too, and hope that it is signs of entering a new phase in this highly valuable area of inquiry; this study could help lead the way in that regard. Please address all points individually in your revision and Reponse. We look forward to seeing it.

Reviewer 1 ·

Basic reporting

Additional citations are needed, please see attached review for suggestions.

Experimental design

Tthis paper could benefit from additional experiments conducting a baseline study using the same methodology on taphonomically altered extant skin, and unaltered, desiccated skin. See description in review.

Validity of the findings

The findings are valid, but could be expanded. See attached

Comments for the author

Review of “Integumentary structure and composition in an exceptionally well-preserved
hadrosaur (Dinosauria: Ornithischia)” by Barbi et al.

These authors use SEM, optical microscopy (OM) and various synchrotron techniques to examine the composition of Late Cretaceous hadrosaur skin recovered from Alberta Canada.

This paper is carefully written and goes to great lengths not to overstate the data presented. That is much appreciated.

The following is a line-by-line commentary on the paper, with conclusions at the end.

Line 52: It would be helpful to note that of these only synchrotron takes these analyses beyond morphology, and then note the limitations of synchrotron in determining the nature and source of the material studied.
Line 62. How do you know that this integument was displaced? The skin, in Fig 1, appears to overlay a rib (patch 2), and patch one appears to originate from the dorsal surface? I am not sure why the assumption of displacement? Also, there is an abbreviation (Ra) on the diagram that is not listed in the figure legend. Additionally, “close-up” is not a technical term, better to use “higher magnification”. Even though “precise mapping” (line 77-78) is impossible, it looks like the specimen is somewhat disarticulated and displaced before the rock falls occurred, thus probably before internment. This seems to suggest the specimen was exposed for some time at the surface before burial. How might this affect your interpretation?
Line 107, was this material studied embedded in a resin before sectioning? Polymers such as these should be compensated for, as they react strongly in vibrational mode analyses. Was the same analysis under the same parameters conducted on a sample of the resin only?
Line 121, please define “pristine skin”. How was this collected? Were gloves worn in the field to prevent transfer of human compounds such as sweat, skin flakes or keratin?
Line 124, a “…section was manufactured…” Standard petrographic sections require that the material be embedded in a polymer resin of some type. What resin was used, under what conditions, and did this resin undergo the same analyses as the skin sample?
Line 130-“pristine piece….embedded in matrix…” if the skin was embedded it was chemically treated, thus no longer “pristine”, and was the matrix in which it was embedded sediment or polymer?
Line 136, please include molarity (or normal) values for HCL used? These authors may have better luck using HF on such skin samples (see Pan et al 2019).
Line 139, please provide more detail? What type of water? What temperature? How were the flakes collected from the water bath?
Line 154-55, how were these parameters determined? Is this standard use? If so, please cite.
Line 168-69, please provide a citation for the use of threshold masking.
Line 181-185, please label the area of sediment in figure 2A using a thin line. In fig 2C, can you outline a ‘scale’ using dashes? Is this a scale with a hinge region as in extant ‘reptiles’? or is the ‘scale’ inferred by the undulating and layered nature of the dark regions?
Line 201-202. There is experimental and other evidence for the role of clays and preservation of fossil components. See McMahon et al 2016, and references therein.
Line 206, fig 4A. this image is blurry (in reviewer copy) and it is hard to detect the “chain of substructures” mentioned in the legend.
Line 215 and figure 5 legend needs citations for these values.
Line 219-220 needs a citation.
Line 223, what “other peaks” of the alcohol group are not present? Please give example and citation.
Line 224 needs a citation, also line 227, 230, 231, 235.
Line 238, please clarify. If thet sample was embedded in polymer to section to required thickness that is a LOT of chemical manipulation.
Line 240, ‘more intriguing’ is your opinion. ‘suggestive’ may be a better choice.
Table 1, citations should be provided for these reference data.
Line 241. Please expand the legend for figure 7.
Line 247 needs a citation, and please clarify why ‘in this case’ the peak at 1789 “would correspond to….” If this peak tends to shift, or change in interpretation, please clarify this.
Line 251, the use of a microtome implies it was embedded in a resin. Please state how the vibrational effects of the resin were compensated for.
Line 254-258, this discussion is a bit confusing, as proteins, lipids and polysaccharides are very different biomolecules, but all are organic. Yet, there was only a “single organic carbon spectrum”?
Line 260-261 need citations for these values and their subsequent assignments.
Line 261-162, I don't think you can rule this out completely using this one method, particularly when these spectra are not compared to biomolecules altered through diagenesis, or through actualistic experiments. A good control would be to also analyses check humics complexed to proteins, or heat treated samples (as a supposed proxy for time)? Baseline data of extant archosaur skin subjected to taphonomic alteration would be a good control.
Line 263-65, is there a precedence for this approach? Please cite.
Line 282-83 needs a citation
Line 310-312, please explain how Fe III and Fe II can coexist in the samples measured here can co-exist, given that one requires oxidizing conditions; the other reducing? Particularly as in Fig 12 B and C, the Fe II and Fe III signals map to the same small fragment at the bottom of the image? What environmental conditions lead to this co-localization of this species? Also, can you please go into more detail as to the meaning of the “featureless spectra”? how is this informative?
Line 319, again please state if these were embedded samples and how interference of resin signal was accounted for. Are these the structures later referred to as “subcircular substructures resembling cells”? How were iron microcrystals ruled out as a source?
Line 332-334, I think that fig 14 and 15 could easily be combined. Figure 15B needs a scale bar.
Line 338, “resembling cells” these structures are too small to be vertebrate cells, and they are the wrong shape to be skin cells. They may, however, be remnants of microbial cells, but the methods employed herein would not allow differentiation.
Line 354, the authors eliminate ‘reandom processes induced by diagenetic modifications” to explain these round substructures. Please specify which “random processes” these might be, and how you are able to eliminate these. If these are remnants of microbial growth on degrading skin, this would be part of diagenesis per se. And, if microbial overgrowth occurred, it is possible that this preserved at some level the skin architecture, even if in replication by biofilm.
Line 359, these substructures are not consistent with skin cells. Again, if they are about 2 micrometers in diameter, this is too small, as most skin cells are quite a bit larger, no matter the taxon. Furthermore, the outer skin layers are dominated by squamous cells fused by tight junctions. These cells are greatly flattened, not subcircular. I agree with the authors in line 357 that these materials are likely associated with the diagenetic processes that led to preservation.
Line 361 should be ‘implicated’
Line 365-66 please support this statement with a citation; also the following.
Line 381-82, there is abundant literature on the correlation between clays and delayed degradation; in part due to the small grain size and high charge of clay particles inactivating bacterial degradative enzymes. This might be helpful to cite.

Supplemental:
Please expand the legends for fig S7 and S8 to indicate to what each color refers. Additionally, in the legend for S8, it is stated that there is an “absence of any remarkable peaks….between 1500 and 1800 cm-1, but to me there appears to be many peaks at 1500 that are significantly above baseline. More explanation is needed for these legends.

General comments
In general, I applaud the authors on the thorough analyses and the use of multiple unrelated techniques to support the general ‘story’ of preservation,—even though these methods did not, in the long run, shed light on the nature of the ‘organic rich layer’, its source (endogenous or exogenous), or ultimately, taphonomic processes to which these materials were exposed. Although this study demonstrates the capabilities of the methods employed, I am not sure exactly what the take-away points are. Is the skin ‘organic’? Apparently this is supported through the identification and localization of carbonyls, but further information is needed to determine the nature of this layer.

This paper is interesting and has potential for the study of originally soft tissues in the fossil record, but it appears incomplete. It would be greatly strengthened by adding a baseline study on experimentally altered extant skin (e.g. high heat, pressure, desiccation, buried in sediments infiltrated with carbonate-saturated waters, to name a few) for comparison. It would also be aided by a similar study on extant unaltered skin (e.g. shed skin from a lizard?).

To the authors’ great credit, they were cautious in interpretation, and did not overstate the data, which is rare.

I recommend publication if the above issues are addressed.

Reviewer 2 ·

Basic reporting

In my opinion, the study satisfies the criteria for basic reporting.

The paper uses very clear and moderated language, without superlatives or hyperbole.

Experimental design

This is a welcome contribution to advanced analytical techniques applied to paleontological material. The research question is well defined and states a clear knowledge gap.

My expertise is in synchrotron radiation applied to paleontological material, but I am not too familiar with the particular synchrotron experiments used here. Nevertheless, the methods are described with a clarity and confidence with the appropriate information required for synchrotron based experiments.

The set of experiments are complimentary and support each other nicely.

Validity of the findings

It was refreshing to read a study where the conclusions were not over inflated.
The suite of techniques used is novel and corroborating.

In fact there is almost no speculation whatsoever, but this is ok as I believe that the results would not support them.

Comments for the author

Line 45: Edwards et al. 2011 applied IR and synchrotron radiation techniques to 50 mya squamate skin from the Green River Formation, but this study is not referenced and probably should be.

Line 191 main text and section 2 of the SI:
I am skeptical of suggesting specific minerals based elemental analysis as many minerals have similar elemental components but different stoichiometry. While these minerals are the most likely candidates and an educated guess based on the authors knowledge of the sample and its geological context, it is misleading to suggest mineral identification this way. Corroborating petrographic or X-ray diffraction for example is far more definitive. Barite is pretty diagnostic from its elemental composition and is lucky that the authors have areas of almost pure barite. Most samples will be more heterogenous and the elemental peaks convolved.

Figure 7 and S7:
IR measurements support the mineralogical assignments made earlier from the elemental analysis. However, there are still some potential ambiguities here and I would really like to see a petrographic/crystallographic analysis done to confirm these minerals. It is not so much the identification of the specific minerals that I care about really, it’s the speculative assignment based on the data presented.

Figure S7 is not really useful and is just a great mess of spectra. Maybe splitting these up into groups would be more useful, plus identifying where each spectrum came from. Maybe you could do a cluster for each horizontal line of the map. I can see at least 3 if not 4 different spectral groups here that would be good so see more clearly.

I am not too familiar with the techniques in the following section so it is not appropriate for me to comment here

Finally, how were the samples handled and stored after sampling in the field? Sterile scalpels, foil envelopes, sealed viles etc. This information should be added.

---

## Round 0.2 · Minor Revisions

The reviewer is satisfied with revisions bar some final points that all seem easily adopted. Well done. Please make these amendments and submit your final MS. Thanks!

Reviewer 2 ·

Basic reporting

As per the original submission I find the basic reporting satisfactory.

The study satisfies the criteria for basic reporting.

The paper uses very clear and moderated language, without superlatives or hyperbole.

Experimental design

No comment

Validity of the findings

No comment

Comments for the author

Thanks for adding the additional information as requested.

I still am not particularly happy with S7 and S8. There are many medium to small variations in these spectra that I would like to be able to see.

I am not suggesting to take these data out, only that they are not much use (in my opinion) in their current format. I suggest maybe breaking one figure into maybe 2-4 panels to help compartmentalize the spectra. Right now I do not feel the figures perform the function that the authors intended.

One last thing that I failed to catch last time a reference to the technical specifications of the beam lines used would be great (sorry if I missed it). This would avoid adding more detail in that regard into this paper such as flux etc.

The additional information the authors have added to the manuscript since the initial submission is welcome.

---

## Round 0.3 · accepted · Accept

Thank you for adding the extra aspects to the paper- it has strengthened as a result. Congratulations on acceptance!